# *Fusobacterium nucleatum* and Malignant Tumors of the Digestive Tract: A Mechanistic Overview

**DOI:** 10.3390/bioengineering9070285

**Published:** 2022-06-28

**Authors:** Yue Lai, Jun Mi, Qiang Feng

**Affiliations:** 1Department of Human Microbiome, School and Hospital of Stomatology, Cheeloo College of Medicine, Shandong University & Shandong Key Laboratory of Oral Tissue Regeneration & Shandong Engineering Laboratory for Dental Materials and Oral Tissue Regeneration, No. 44-1 Wenhua Road West, Jinan 250012, China; ly2882881@163.com; 2Shenzhen Research Institute of Shandong University, Shenzhen 518057, China

**Keywords:** *Fusobacterium nucleatum*, oral squamous cell carcinoma, colorectal carcinoma, esophageal squamous cell carcinoma, pancreatic carcinoma

## Abstract

*Fusobacterium nucleatum* (*F. nucleatum*) is an oral anaerobe that plays a role in several oral diseases. However, *F. nucleatum* is also found in other tissues of the digestive tract, and several studies have recently reported that the level of *F. nucleatum* is significantly elevated in malignant tumors of the digestive tract. *F. nucleatum* is proposed as one of the risk factors in the initiation and progression of digestive tract malignant tumors. In this review, we summarize recent reports on *F. nucleatum* and its role in digestive tract cancers and evaluate the mechanisms underlying the action of *F. nucleatum* in digestive tract cancers.

## 1. Introduction

Results from a number of studies have shown that there are different microbiota in the tumor environment, and intestinal flora is a major factor involved in cancer mechanisms [1,2]. Therefore, there may be specific carcinogenic bacteria that initiate or promote cancer.

*F. nucleatum*, a Gram-negative bacterium, is a normal component of human oral microecology. *F. nucleatum* exists not only in the human oral cavity, but also in other tissues of the digestive tract, such as the gastrointestinal tract. It was firstly discovered in patients with periodontal disease and was considered as a potential periodontal pathogen [3]. As a permanent member of the oral microbiota, *F. nucleatum* binds to abiotic surfaces, host cells, or other microorganisms, mainly through the mediating effect of the adhesion factors. Moreover, as a bonding bridge between early and late pathogens located on the surface of the teeth and epithelium [4], *F. nucleatum* plays an important role in the development of periodontal disease [5]. *F. nucleatum* is an opportunistic pathogen, not only involved in inflammatory processes, such as periodontitis [6], inflammatory bowel disease [7], pancreatic abscess [8], premature birth [9], and liver abscess [10], but also involved in the progression of cancer, which include oral squamous cell carcinoma (OSCC), colorectal carcinoma (CRC), esophageal squamous cell carcinoma (ESCC), pancreatic carcinoma (PC), gastric carcinoma (GC), liver carcinoma (LC), and breast carcinoma [11]. Among them, *F. nucleatum* is considered to be the main cause of CRC and PC [12].

Malignant tumors of the digestive tract are present in the esophagus, stomach, liver, pancreas, colon, and rectum. Their morbidity and relative mortality rates are increasing worldwide [13]. This paper reviews recent studies on the association and possible mechanisms between *F. nucleatum* and digestive tract cancers, in order to understand the relationships and contribute to future related research studies.

## 2. Relationships between *F. nucleatum* and Malignant Tumors of the Digestive Tract

### 2.1. F. nucleatum and OSCC

OSCC is the most common malignancy in the oral cavity and is considered to be the main cause of death due to oral diseases in many countries. Currently, recognized risk factors include smoking, drinking, chewing betel nut, etc., but 15% of OSCCs cannot be explained by these factors [14]. Continuous explorations of researchers indicate that *F. nucleatum* may be involved in the development of OSCC.

In 1998, Nagy et al. found that the levels of *F. nucleatum* and *Porphyromonas gingivalis* (*P. gingivalis*) were significantly higher in OSCC than in normal tissue [15]. Then, some researchers sequenced DNA and found that the level of *F. nucleatum* in OSCC-lesion surface swabs was significantly higher than that in normal mucosa from the same patients [16]. The results of comparative analyses showed that bacterial biomarkers were associated with OSCC, and the most distinct genera were *F. nucleatum* (enriched in OSCC) and *Streptococcus* (reduced in OSCC) [17]. The results of a metagenomic analysis found that the abundances of *F. nucleatum* and some other bacteria were significantly increased in the patients compared with the controls [18]. Meanwhile, *F. nucleatum* was found to play an important role in the progression of OSCC. Kang W et al. suggested that *F. nucleatum* infection in the oral cavity has a potential tumor-promoting effect [19]. Moreover, some further studies showed that it could protect tumor cells from immune cell attack, stimulating the progression of OSCC through the toll-like receptor (TLR) contact with the oral epithelium [20,21]. Besides, *F. nucleatum* levels in OSCC were significantly negatively associated with B-lymphocyte, CD4^+^T-helper-lymphocyte, M2-macrophage, and fibroblast markers, indicating that *F. nucleatum* plays a role in the host anti-tumor immune reaction [14]. Researchers found an association between *F. nucleatum* and high cytokine levels in CRC and OSCC; thus, it could generate pro-inflammatory factors through the lipopolysaccharide (LPS) activation of the TLR4-mediated nuclear factor-κB (NF-κB) signaling pathway in the outer membrane to create an inflammatory microenvironment and promote tumor progression [22]. *F. nucleatum* could promote OSCC cell proliferation by up-regulating cyclin D1 and c-myc [23]. These studies suggest that *F. nucleatum* may play a vital role in the progression of OSCC through different approaches, which needs further research.

### 2.2. F. nucleatum and CRC

Globally, CRC is the third most prevalent malignancy and the second most frequent cause of death from malignancies. According to the GLOBOCAN project of World Health Organization (WHO) Cancer Research Center, the incidence of CRC worldwide was about 1,880,725 in 2020, and mortality was about 915,880 [24]. Studies found a close relationship between *F. nucleatum* and CRC. Elevated levels of *F. nucleatum* were detected in the colon tissues of CRC patients in comparison with healthy people [25]. The results of a meta-analysis showed that CRC patients with high tissue abundance of *F. nucleatum* had poor survival rates [26]. Furthermore, many studies suggested that the overabundance of *F. nucleatum* may be associated with CRC carcinogenesis. The results of a metagenomic analysis found that *F. nucleatum* proved to be an important marker of colorectal carcinogenesis and tumor aggressiveness [27]. A co-culture of *F. nucleatum* and CRC cells could increase formate secretion and cancer glutamine metabolism, which drove CRC tumor invasion and proliferation by triggering aryl hydrocarbon receptor (AhR) signaling [28]. *F. nucleatum* was associated with the elevation of angiopoietin-like 4 protein (ANGPTL4) expression in CRC cells, thus increasing glycolytic activity, which plays an important role in *F. nucleatum* colonization and proliferation in CRC [29]. After investigating the clinicopathological features and prognostic impact of *F. nucleatum* status in patients with CRC, researchers found that a greater amount of *F. nucleatum* had a significant association with low-level microsatellite instability (MSI)/microsatellite stability (MSS), which indicates that *F. nucleatum* might be associated with poor prognostic [30]. *F. nucleatum* in CRC could selectively expand immunosuppressive myeloid cells to form an immunosuppressive tumor microenvironment, thus inhibiting T-cell proliferation and inducing T-cell apoptosis in CRC [31]. These studies suggest that *F. nucleatum* may play an important role in the progression of CRC through different approaches, which needs further research.

### 2.3. F. nucleatum and ESCC

Studies showed a potential association between *F. nucleatum* and ESCC. A retrospective study found that the abundance of *F. nucleatum* in ESCC tissues was significantly associated with the pT stage and clinical stage [32]. In ESCC tissues, the *F. nucleatum* DNA level was higher than that in adjacent non-tumor tissues, and the higher DNA level of *F. nucleatum* was significantly associated with cancer-specific survival, poor prediction of relapse-free survival, and poor response to neoadjuvant chemotherapy [33,34,35]. In addition, some studies suggested that the overabundance of *F. nucleatum* may be associated with ESCC carcinogenesis. Studies showed that *F. nucleatum* invaded ESCC cells and induced the NF-κB pathway through the nucleotide oligomerization domain 1 (NOD1) and receptor-interacting protein kinase 2 (RIPK2) pathways, leading to tumor progression [36]. *F. nucleatum* infection could induce a high expression of NOD-like receptor protein 3 (NLRP3) in ESCC, thus leading to myeloid-derived suppressor cell (MDSC) enrichment and weakening the body’s antitumor immunity [37]. Additionally, *F. nucleatum* infection and colonization induced a high expression of KIR2DL1 on the surface of CD8^+^T cells, which weakened the antitumor immune response and promoted the malignant progression of ESCC [38]. Currently, there are few studies on the mechanisms of ESCC induced by *F. nucleatum*, which are worthy of further research.

### 2.4. F. nucleatum and PC

PC is the third cause of cancer death in the United States and the seventh cause of cancer death worldwide [24,39]. Recent studies showed that the development of PC may be associated with *F. nucleatum*. Studies found that the co-occurrence and enrichment of *F. nucleatum* in cyst fluid from intraductal papillary mucinous neoplasms (IPMNs) with high-grade dysplasia made IPMNs progress to invasive PC and decreased PC patients’ survival [40]. A cross-sectional study found that the cancer-specific mortality rate in the positive group was significantly higher than that in the control group according to a multivariate Cox regression analysis, which suggested that *F. nucleatum* is independently associated with poor prognosis of PC and that it might also be a prognostic biomarker of PC [41]. Nowadays, there are few studies on the mechanisms of PC induced by *F. nucleatum*, which are worthy of further research.

### 2.5. F. nucleatum and GC

GC is the fourth leading cause of cancer death worldwide [24], and its main risk factor remains *Helicobacter pylori (H. pylori)*, which is known to be associated with 90% of GC cases [42].

A potential association between *F. nucleatum* and GC was found in many studies. A case-control study showed that *F. nucleatum* increased cancer risk factors [43]. Some researchers evaluated the possible association between the abundance of some periopathogens in the subgingival plaque and periodontal status and the characteristics of gastric cancer, and the results showed that the most abundant bacteria were *F. nucleatum* followed by *T. forsythia* in all groups [44]. Moreover, others also found that *F. nucleatum*-positive GC patients had significantly worse overall survival (OS) than the control group [45]. A cohort study on the relationship between *F. nucleatum* and *H. pylori in* GC showed that *F. nucleatum* colonization led to poor prognosis in patients with advanced *H. pylori*-positive GC [46]. However, there are few studies about *F. nucleatum*’s role in GC, which needs further research.

### 2.6. F. nucleatum and LC

As a malignant cancer with high morbidity and mortality [47], LC poses a great threat to the health of people all over the world. Lu H et al. identified *F. nucleatum* as a possible biomarker for identifying patients with LC, showing that there were significant differences in the relative abundance between healthy controls and patients with LC [48]. However, there are few studies of *F. nucleatum*’s role in LC, which is worthy of further research.

## 3. Underlying Mechanisms of Action

As noted above, several studies showed a significant relationship between the frequency of *F. nucleatum* and digestive tract cancers, but there are few studies available on this issue, and the responsible mechanisms have not yet been well defined. In this section, we review studies on the mechanisms underlying the action of *F. nucleatum* in the tumorigenesis of these tumors and the classification of factors (Figure 1).

### 3.1. Bacterial Adhesion and Colonization

*F. nucleatum* is abundant in the oral cavity and intestinal tract, and tumor cells can attract *F. nucleatum* adhesion and colonization through specific epigenetic modifications. It was shown that a high load of *F. nucleatum* was associated with specific epigenetic phenotypes of CRC. Some researchers showed that *F. nucleatum* was associated with some molecular changes in CRC, such as CpG island methylation phenotype (CIMP), TP53 wild type, Human Mutl Homolog 1 (hMLH1) methylation, MSI, and CHD7/8 mutation [45]. However, the mechanisms are still poorly defined. It was found that these specific molecular characteristics of CRC mainly occur in the ascending colon, which is the most common colonization site of *F. nucleatum* in the gastrointestinal tract [49]. This may indicate that there is a certain relationship between *F. nucleatum* and the colonic mucosal microenvironment. *F. nucleatum* can colonize digestive tract tissues and organs by direct diffusion under the attraction of tumor cells [50]. Intestinal permeability in patients with periodontitis is enhanced by the chemotactic substances produced and secreted by plaque microorganisms and host cells. *F. nucleatum*, an important periodontal pathogen, can enter target tissues through paracellular pathways [51]. In addition, it can enter target tissues via circulatory (and hematological or lymphatic) pathways [52,53].

After reaching the surface of the target tissue, *F. nucleatum* binds to the target tissue through Fusobacterium adhesin A (FadA), Fusobacterium autotransporter (Fap2), and other adhesions present on the surface (Pathway 1 in Figure 1). *F. nucleatum* infects digestive epithelial cells or tumor cells and binds to the 11-AA domain of the E-cadherin EC5 domain via its specific adhesive FadA, which is internalized by E-cadherin [54]. The 11-AA inhibitory peptide is known to inhibit the binding and invasion of *F. nucleatum* and eliminate all subsequent host responses, including tumor growth and inflammatory responses [55]. Therefore, FadA inhibits the function of the 11-AA inhibitory peptide by binding to the 11-AA inhibitory peptide and activates the β-catenin signaling pathway, resulting in an increased expression of transcription factors, oncogenes, Wnt genes, and inflammatory genes, and in the up-regulation of the NF-κB and Wnt pathways, promoting the proliferation of CRC cells. Fap2 on the surface of *F. nucleatum* interacts with the over-expressed D-galactose-β (1-3)-N-acetyl-D-galactosamine (Gal-GalNAc) lectin sugar portion of tumor cells to colonize, multiply, and survive in tissues, thus further promoting tumorigenesis [56]. In addition, *F. nucleatum* promotes the colonization of other bacteria on the surface of target tissues by participating in the formation of biofilms, which can cause the inflammation of target tissues, increasing the permeability of pancreatic epithelium and allowing bacteria and their harmful products to enter deep tissues (Pathway 1 in Figure 1).

### 3.2. Activation of Tumor Cell Proliferation

Altered energy metabolism, a biochemical fingerprint of cancer cells, represents one of the “hallmarks of cancer”. This metabolic phenotype is characterized by preferential dependence on glycolysis (the process of conversion of glucose into pyruvate followed by lactate production) for energy production in an oxygen-independent manner. Recently, some researchers found that *F. nucleatum* could activate tumor cell proliferation by the activation of glycolysis. Hong J et al. found that *F. nucleatum* activated glycolysis and carcinogenesis via a selective increase in long non-coding RNA (lncRNA) enolase1-intronic transcript 1 (ENO1-IT1). Moreover, through the 3′ fragment of ENO1-IT1 mediating the interaction with keratin 7 (KAT7), ENO1-IT1 coordinates the acetylation of histones genome-wide. This contributes to regulating enolase1 (ENO1) transcription in CRC via epigenetic modulation [57]. Because ENO1 is a key glycolytic enzyme that catalyzes the conversion of 2-phosphoglycerate to phosphoenolpyruvate (PEP), *F. nucleatum* can promote CRC glycolysis via ENO1 [58]. Moreover, glycolysis serves as the main source of energy metabolism, and *F. nucleatum* can promote CRC glycolysis through ENO1, which contributes to tumor cell proliferation (Pathway 2 in Figure 1).

Meanwhile, some studies found that *F. nucleatum* infection promoted tumor cell proliferation through the Ku70/p53 pathway [59]. Under normal circumstances, DNA double-strand break (DSB) is formed through homologous recombination or is repaired by Ku70 and Ku80. Once cells are damaged, the DNA-binding domain of Ku70 is acetylated, while the Ku70-dependent suppression of p53 expression is abrogated. That is, the expression of Ku70 is reduced, while the expression of p53 is increased [60,61]. However, when *F. nucleatum* infection occurs, Ku70 expression is decreased, which can lead to the down-regulation or even abnormal mutation of p53 expression in association with enhanced cell proliferation abilities [59,62]. The authors hypothesize that the DNA damage caused by *F. nucleatum* is so serious that the intracellular Ku70 is insufficient to provide prompt repair. Furthermore, without the normal regulation of Ku70, wild p53 is aberrantly expressed from up-regulation to down-regulation along with a possible mutation that may enhance cell proliferation abilities. The Ku70/p53 pathway inhibits the p53-mediated post-injury response, which enhances the proliferation of OSCC cells [63] (Pathway 2 in Figure 1).

### 3.3. Fusobacterium Promoting Digestive Tract Cancers Development from Inflammation to Malignancy

The causation of cancer by *F. nucleatum* could occur in cases where the bacteria are involved in chronic inflammation. Inflammasome is a multi-protein complex composed of pattern recognition receptor (PRR), apoptosis-associated speck-like protein containing CARD (ASC), and caspase1 [64], which mediates the host immune system to response to microbial infection and cell damage. A large number of clinical samples showed that patients with periodontal diseases had a significantly increased possibility of oral and gastrointestinal cancers [65,66]. As one of the common periodontal pathogens, *F. nucleatum* promotes cytokine production and adjusts the environment to a more pro-inflammatory state, stimulating chronic inflammation and possibly promoting tumor proliferation [67,68]. By driving the expression of higher levels of cytokines (TNF-α, IL-6, IL-8, and IL-1β) in inflammatory tissues through the NF-κB pathway, *F. nucleatum* promotes the aggregation of tumor-associated neutrophils (TANs) and tumor-associated macrophages (TAMs), producing nitric oxide (NO) in inflammatory sites. Among these expression signatures, IL-8, TNF-α, and other chemokines could recruit neutrophils and macrophages, which synthesize nitric oxide (NO) and cause oxidative stress to epithelial and stromal cells. This results in DNA damage and the consequent activation of p53 transcription, which, in turn, suppresses tumorigenesis by inducing G1-S arrest, DNA repair, and cell apoptosis. Moreover, p53 overexpression also leads to TP53 mutation, which is a key event during CRC development [69]. Besides, *F. nucleatum* can promote the expression of absent in Melanoma (AIM2) inflammasome, which can up-regulate the expression of IL-1β and down-regulate the expression of pyrin domain (PYD)-only protein 1 (POP1), thus regulating NLR family pyrin domain containing 3 (NLRP3) inflammasome activation by targeting ASC [70] (Pathway 2 in Figure 1). In addition, it is evident that there may be a close relationship among periodontal bacterial infection, periodontitis, and oral squamous cell carcinoma. In other words, there may be a periodontal bacterial infection–periodontitis–OSCC association and regulatory mechanism [71]. *F. nucleatum* infection activates the upstream signaling molecules of ATR-CHK1 and inhibits CHK1 activation, which promotes the over-expression of NLRP3 and cell proliferation, inhibiting apoptosis, thus promoting OSCC cell survival (Pathway 3 in Figure 1).

### 3.4. Cell Migration and Invasion

The epithelial–mesenchymal transition (EMT) is defined as a change from the epithelial to mesenchymal phenotypes that is usually rapid and reversible, often accompanied by weakened cell–cell junctions and the remodeling of the cytoskeleton [72]. Many studies showed that partial EMT was associated with cancer progression [73]. Recently, some researchers found that *F. nucleatum* could promote cell migration and invasion through EMT. *F. nucleatum* infects normal or cancerous oral epithelial cells, inhibiting the expression of miR-296-5p and SNAIL by up-regulating lncRNA miR4435-2HG, while miR-296-5p further negatively or indirectly down-regulates SNAIL expression by Akt2. Therefore, *F. nucleatum* triggers EMT to promote OSCC cell migration through the lncRNA miR4435-2Hg/miR-296-5p/Akt2/SNAIL signaling pathway. However, it does not promote cell proliferation or cell cycle progression [74] (Pathway 3 in Figure 1). In addition, E-cadherin is down-regulated and trans-located to the cytoplasm after *F. nucleatum* infects normal or cancerous oral epithelial cells. N-cadherin, Vimentin, SNAIL, matrix metalloprotease-2 (MMP-2), and Toll-like receptor 4 (TLR-4) are up-regulated, and TLR-4 is responsible for LPS recognition. LPS may up-regulate p-EMT through the TLR signaling pathway, which changes the gene expression in OSCC and transforms epithelial cells into a p-EMT phenotype, thus promoting OSCC cell migration [75].

Matrix metalloproteinase-13 (MMP-13), also known as collagenase-3, plays a key role in normal metabolism and homeostasis. In addition, it also plays an important role in inflammatory response, tumor invasion, and metastasis [76,77]. Some studies showed that *F. nucleatum* could promote cell migration and invasion through the enzymatic degradation of the extracellular matrix. *F. nucleatum* infects oral epithelial cells and produces a large number of cytokines, such as transforming growth factor-α/β (TGF-α/β) and keratinocyte growth factor (KGF), which promote MMP-13 production by activating p38 MAP kinase [78]. MMP-13 lyses laminin-5 and transforms it into a form that promotes cell migration, thus promoting the invasion and metastasis of OSCC [54]. In addition, *F. nucleatum* can degrade the extracellular matrix and destroy the physical barrier by the secretion of MMP-2, MMP-9, and other MMPs stimulated by epithelial cells, facilitating the invasion and migration of OSCC [79] (Pathway 4 in Figure 1).

Some researchers also found that *F. nucleatum* can induct autophagy, which can promote cell migration and invasion. Yu et al. found that *F. nucleatum* could activate the autophagy pathway of CRC. The pathway was proved to inhibit the occurrence and development of tumors by mainly inhibiting the migration and invasion of tumor cells in vitro and by weakening metastasis in vivo [80,81]. Caspase activation and recruitment domain 3 (CARD3, RIP2) is a serine/threonine/tyrosine kinase with a carboxy-terminal caspase activation and recruitment domain (CARD). The abundance of *F. nucleatum* was found to be positively associated with CARD3 expression [82]. The down-regulation of CARD3 expression can reduce the migration, autophagosome formation, and expression of autophagy-associated proteins, which are induced by *F. nucleatum* infection. Therefore, the infection of *F. nucleatum* with CRC cells increases the migration of cancer cells; up-regulates the expression of CARD3, LC3-II, Beclin1 and Vimentin; and down-regulates the expression of E-cadherin and P62 in CRC cells. Through this mechanism, *F. nucleatum* specifically targets CARD3 to activate autophagy and promote CRC migration. In addition, *F. nucleatum* also stimulates the expression of pULK1 autophagy-related proteins, ULK1, and ATG7 by the loss of miR-18a/4802 and the reliance on the TLR4 and MYD88 signaling pathways, thus activating autophagy to promote CRC migration mechanism. Besides, it is also closely related to drug resistance in CRC [80] (Pathway 3 in Figure 1).

Studies showed that *F. nucleatum* significantly up-regulated the expression of lncRNA keratin 7-antisense (KRT7-AS) and KRT7 in CRC by activating the NF-κB pathway (Pathway 3 in Figure 1). KRT7, a type II cytokeratin, is a component of the cytoskeleton and epithelial intermediate filaments [83]. In addition to maintaining the integrity of the cell structure, it can also promote cell motility [84]. Bayrak R. et al. found that KRT7 was more common in CRC with lymph node metastasis than in CRC without lymph node metastasis [85]. Subsequently, KRT7 acts as a downstream target of KRT7-AS, which further promotes CRC cell migration through the up-regulation of KRT7 [86].

Researchers found that *F. nucleatum* could promote cell migration and invasion by secreting exosomes (Pathway 3 in Figure 1). For example, Guo S et al. found that *F. nucleatum* could promote CRC migration by stimulating the production of Mir-1246/92B-3p/27A-3p and CXCL16/RhoA/IL-8 exosomes in CRC cells [87].

There is a specific association between bacteria in dental plaques. *F. nucleatum* is the bonding bridge between early and late colonization bacteria, playing a leading role in the late stage of plaque biofilm formation [88]. Studies showed that *F. nucleatum* may also synergistically promote the cell migration and invasion of OSCC with other oral bacteria, such as *P. gingivalis* (Pathway 3 in Figure 1). For example, in the interaction between *F. nucleatum* and *P. gingivalis*, they infected oral epithelial cells and directly interacted with oral epithelial cells through the TLR signaling pathway to produce IL-6. The activation of signal transducer and activator of transcription 3 (STAT3) activated Cyclin-D1, MMP-9, heparin, etc., which promote the proliferation and invasion of OSCC cells [22].

### 3.5. Producing a Tumor Immunosuppressive Microenvironment

*F. nucleatum* induced by Fap2 mediates T-cell immunoglobulin and ITIM domain (TIGIT) or promotes the apoptosis of lymphocytes and NK cells by activating carcinoembryonic antigen-related cell adhesion molecules 1 (CEACAM1), and protects CRC cells from the cytotoxic effects of NK cells and T lymphocytes [18,89]. Meanwhile, Fap2 is a galactose-binding lectin on the surface of *F. nucleatum*, and its ligand, Gal-GalNAc, is on the surface of CRC tumors [56]. Thus, Fap2 can bind to Gal-GalNAc and mediate the recruitment of *F. nucleatum* to CRC cells, enhancing the role of *F. nucleatum*. In addition, researchers measured *F. nucleatum* DNA in the tumor tissues of 933 of 4465 CRC cases (including 128 *F*. *nucleatum*-positive cases) in two prospective cohorts by qPCR, and the results showed that *F. nucleatum* was negatively associated with tumor matrix CD3^+^T lymphocytes [90]. Kostic et al. confirmed that *F. nucleatum* promoted CRC by inhibiting proliferation and inducing T-cell apoptosis [91]. In addition, *F. nucleatum*-secreted autoinducer-2 (AI-2) acted on the tumor necrosis factor ligand superfamily member 9 (TNFSF9) signaling pathway to reduce CD4^+^T cells/CD8^+^T cells in CRC tissues, influencing the progression of CRC [92] (Pathway 5 in Figure 1).

## 4. Conclusions

In summary, many studies have shown that *F. nucleatum* is associated with OSCC, CRC, ESCC, PC, GC, and LC. The questions of whether *F. nucleatum* can cause disease alone or whether there are related co-mechanisms factors and how they function need to be further explored. In addition, current studies have mostly focused on the association between *F. nucleatum* and malignant tumors of the digestive tract, and there are still many gaps in etiology, mechanisms, immunology, and other aspects among them. In terms of other digestive tract cancers, such as LC, only a few studies have speculated on the association between them, which has not been supported by a large number of studies.

## Figures and Tables

**Figure 1 bioengineering-09-00285-f001:**
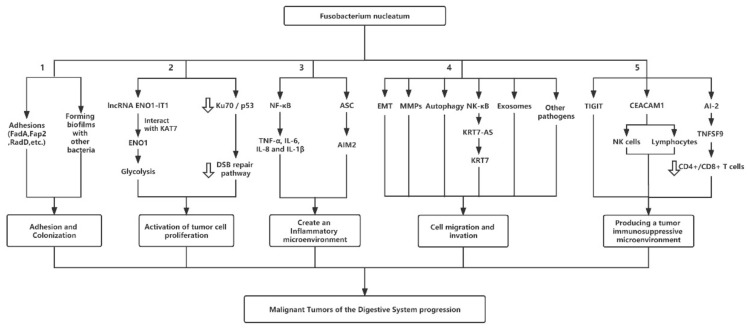
Underlying mechanism of *Fusobacterium nucleatum* pathogenesis in digestive tract cancers. FadA, Fusobacterium adhesin A; Fap2, Fusobacterium autotransporter; ENO1-IT1, enolase1-intronic transcript 1; KAT7-AS, keratin 7-antisense; KAT7, keratin 7; ENO1, enolase1; DSB, DNA double-strand break; NF-κB, nuclear factor-κB; ASC, apoptosis-associated speck-like protein containing a CARD; AIM2, absent in Melanoma; EMT, epithelial-mesenchymal transition; MMPs, matrix metalloproteinases; TIGIT, T-cell immunoglobulin and ITIM domain; NK, natural killer; CEACAM1, carcinoembryonic antigen-related cell adhesion molecules 1; AI-2, autoinducer-2; TNFSF9, tumor necrosis factor ligand superfamily member 9.

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
