# Peer review of "Fusobacterium nucleatum and Malignant Tumors of the Digestive Tract: A Mechanistic Overview"

_bioengineering, 2022, doi:10.3390/bioengineering9070285_

Round 1
Reviewer 1 Report
Well written review article. However, oral cavity (though technically the start of the digestive tract) is not considered as a part of the digestive system.
So, either references to OSCC including the mechanisms have to be removed or title modified accordingly
Reviewer 2 Report
The authors present findings linking the oral anaerobic bacterium, F. nucleatum, to tumorigenesis in the digestive tract. The first few sections highlight the need for further work based on scant evidence linking the bacterium to cancer in each section of the digestive tract or conflicting findings, particularly with promoting or suppressing inflammation.
In some sections poor phrasing or incomplete explanations make the work hard to follow. In section 2.2, please spell out what the AhR pathway is and rewrite the sentence on ‘low microsatellite instability', as on reading the original paper I found this statement to be misleading. At the end of that same sentence the word “effect” is not proper.
In sections 2.4 and 2.5, both PC and GC are described as the 3rd leading/highest cause of cancer deaths. Which is it?
The words, association and correlation are used as if they represent the same thing. “Association refers to any relationship between two variables, whereas correlation is often used to refer only to a linear relationship between two variables” (Statistical Reference guide). Generally, they can be switched. However, here they should be used more carefully.
There are several grammar issues. Please check the text using an English grammar/spell check program or a technical service. Eg. 3.1- F. nucleatum can be colonized. Section 3.2- Due to ENOS1 is a key; “(significantly) increased the possibility--------cancers sigficicanlty.
In section 3.2, please explain how absence of Ku70/p53 with resulting failure to repair DSB results in stimulation of proliferation. The same for NO, after allowing DNA damage.
Section 3.3 last paragraph on page – F. nucleatum can induce autophagy. The abundance of F. nucleatum was “found” to be ----, not proved. Proved is used above this, where the word, ” found “ would be more appropriate.
Section 4. Causation of cancer by F. nucleatum could occur is cases where the bacteria are involved in chronic inflammation, which might occur in OSCC and in IBD. I don’t see anything here on F. nucleatum and IBD and an explicit link between OSCC and chronic inflammation is not spell out. There should probably be a separate section on this topic.
Reviewer 3 Report
Fusobacterium nuclatum is implicated in OSCC, CRC, ESCC, PC, GC and LC. The authors describes also Fusobacterium nucleatum as one of the risk factors in the initiation and progression of digestive system malignant tumors .The authors described the bacterium pathogenesis and its correction to digestive malignancy based on an extended bibliography and trying to explain its role as a risk factor.The strain could be a vsaluable bioindicator for the disease .
It is a well written interesting article for the scientific community.
The authors must write in italics the bacterial genus and species in the paper.
My suggestion is to ACCEPT and publish it after this minor correction.
Round 2
Reviewer 2 Report
Looks OK